# Feasibility and Quality Validation of a Mobile Application for Enhancing Adherence to Opioids in Sickle Cell Disease

**DOI:** 10.3390/healthcare10081506

**Published:** 2022-08-10

**Authors:** Daniel M. Sop, Taylor Crouch, Yue Zhang, Thokozeni Lipato, John Wilson, Wally R. Smith

**Affiliations:** 1Department of General Internal Medicine, Virginia Commonwealth University, Richmond, VA 23298, USA; 2Department of Biomedical Engineering, Virginia Commonwealth University, Richmond, VA 23298, USA; 3Department of Psychiatry, Virginia Commonwealth University, Richmond, VA 23298, USA

**Keywords:** medical apps, pain management, opioids, chronic condition, sickle cell disease, mHealth

## Abstract

Prescription opioid nonadherence, specifically opioid misuse, has contributed to the opioid epidemic and opioid-related mortality in the US. Popular methods to measure and control opioid adherence have limitations, but mobile health, specifically smartphone applications, offers a potentially useful technology for this purpose. We developed, tested, and validated the OpPill application using the Mobile Applications Rating Scale (MARS), a validated tool for assessing the quality of mobile health apps. The MARS contains four scales (range of each scale = 0–4) that rate Engagement, Functionality, Aesthetics, and Information Quality. It also assesses subjective quality, relevance, and overall application impact. Our application was built to be a mobile monitoring and reporting system intended to enhance opioid adherence by collecting data and providing systematic feedback on pain and opioid use. Patients (*n* = 28) all had one of various SCD genotypes, were ages 19 to 59 years (mean 36.56), 53.6% were female, and 39.3% had completed some college. Patients rated the OpPill application highly on all four scales: Engagement, 3.93 ± 0.73; Functionality, 4.54 ± 0.66; Aesthetics, 3.92 ± 0.81; Information, 3.91 ± 0.87. The majority of patients found the application to be relevant for their care. A total of 96% reported the information within the app was complete, while 4% estimated the information to be minimal or overwhelming. Patients (91.7%) overwhelmingly reported that the quality of information as it pertained to SCD patients was relevant; only 8.3% found the application to be poorly relevant to SCD. Similarly, patients (91.7%) overwhelmingly rated both the application’s performance and ease of use positively. The large majority of participants (85.7%) found the application to be interesting to use, while 74% found it entertaining. All users found the application’s navigation to be logical and accurate with consistent and intuitive gestural design. We conclude that the OpPill application, specifically targeted to monitor opioid use and pain and opioid behavior in patients with Chronic Non-Cancer Pain, was feasible and rated by SCD patients as easy-to-use using a validated rating tool.

## 1. Introduction

According to the International Society for Pharmacoeconomics and Outcome Research (ISPOR), adherence is “the extent to which a patient acts in accordance with the prescribed interval, and dose of a dosing regimen” [1]. Poor adherence causes approximately 33% to 69% of medication-related hospitalizations and accounts for $100 billion in annual healthcare costs [2]. Irrespective of disease, medication complexity, or how adherence is measured, the average adherence rate to chronic medication therapy is only approximately 50% [3]. Medication nonadherence can adversely affect patient health, negatively impact a patient’s relationship with his/her care provider, skew results of clinical therapy trials, and increase health resource consumption [4,5].

Perhaps nowhere is medication nonadherence more relevant in our society than opioid nonadherence, specifically opioid misuse. It is well-recognized that there is an opioid prescription epidemic in the US. Accordingly, the CDC has issued guidelines that recommend against the use of high-dose opioids [6]. In addition, efforts are underway to promote safe opioid use, improve opioid adherence, and prevent prescription opioid diversion by identifying high-risk patients and by educating patients, as well as families, regarding the safe use, storage, and disposal of opioids.

A significant portion of patients with sickle cell disease (SCD), the most common inherited blood disorder, use prescribed opioids regularly. SCD affects the hemoglobin structure of red blood cells, such that they form a sickle shape when deoxygenated. SCD produces a progressively disabling illness with severe clinical consequences. Symptoms of SCD vary but are highlighted by sudden acute unbearable pain throughout the body, known as crises, in addition to profound, hemolytic anemia. A large descriptive diary study of pain and opioid use in SCD found that short-acting and long-acting opioid use was prevalent [7]. The severe and unpredictable pain characteristic of SCD crises puts patients at risk of insufficient pain management, especially given recent heightened scrutiny on prescription opioid use. Due to the high prevalence of opioid use in patients with SCD, they are often stigmatized as drug-seeking [8]. At the same time, when using the ISPOR definition, many SCD patients exhibit medication nonadherence. In a study aiming to understand adherence to opioids in sickle cell disease, many patients with SCD took their medication differently than clinically instructed during and between painful episodes, both underusing and overusing their opioids [9].

Even though SCD is a unique condition as it relates to pain, in many ways it typifies chronic non-cancer pain. Clinicians who prescribe opioids to patients with chronic non-cancer pain must be concerned about the opioid epidemic and about patient safety, while simultaneously addressing patients’ pain needs.

Currently, popular methods to measure adherence, including patient self-reports, pill counts, refill rates, biological monitoring, and electronic monitoring, have limitations and are only proxy measures [10,11,12]. Patient self-reports rely on memory and are prone to inaccuracies and recall bias [13]. Pill counts are unreliable if patients fail to return bottles or discard pills before the count. Research in sickle cell disease has shown that biological monitoring, such as urine toxicology screens, lack precision in quantifying medication use [9]. The use of electronic monitoring devices that detect the opening and closing of a medication bottle, such as the Medication Event Monitoring System (MEMS), has shown validity [14]; however, it does not reflect direct medication ingestion and could be thwarted by patients attempting to hide overuse or underuse. Currently, the most reliable way to quantify medication adherence are digital pill or ingestible biosensor systems. Although reliable, these systems are currently poorly available, still largely experimental, and too expensive for widespread use [15], especially for SCD care, due to the frequent socioeconomic disparities among these patients.

Mobile health (mHealth), the general term for the use of mobile phones and other wireless technology in medical care (http://www.himss.org/mhealth, accessed on 2 February 2019), is being sought to improve prescribing, adherence, patient safety, and health outcomes. mHealth applications are already widely used for health improvement in other chronic diseases and have shown benefits [16,17,18]. mHealth, specifically smartphone applications, also offer a potentially useful technology to assist providers. In order to safely start, adjust, taper, and stop opioids, clinicians need to better understand contextual opioid adherence, including not only timing and dosage, but also the biopsychosocial and environmental context of patient dosing. Context may include the severity of pain that triggered the opioid use, stress level, other physical or emotional symptoms, and even the weather at the time. All are known to affect pain in SCD [19,20]. This type of technology has been developed and piloted in several sickle cell studies with a focus on pediatric and adolescent patients [21,22,23,24]. Although helpful, translation to application in adult patients has proven difficult. This is exacerbated by increased disease severity in adulthood, including increased risk of mortality due to multiorgan failure, chronic pain, and neurocognitive deficits.

With a better understanding of the opioid use pattern and context, clinicians may better identify pain triggers or exacerbating factors unique to each patient, develop improved individualized opioid management plans, and more intelligently apply non-pharmacologic interventions to mitigate pain and opioid use. Furthermore, with a medicinal services industry seeing nonadherence rates of as high as 50 percent and yearly expenses of between $100 billion and $300 billion, the potential return on investment from utilizing cell phone adherence applications could potentially outweigh the burden of nonadherence. This accessible innovation may offer numerous insights that could assist patients and healthcare providers in improving medication adherence.

### Study Rationale

A systematic review of Internet-based medication adherence interventions found 13 studies, each of which lacked quality measurements of adherence [25]. Various studies of the use of smartphones in the clinical setting have been performed [26], but studies empirically testing smartphone applications’ utility in chronic pain for improving adherence in adults with SCD are lacking. Thus, the goal of our study was to develop and test the acceptability and usability of a mobile software application among adults with SCD to increase adherence to prescribed opioids. Additionally, the application aims to allow patients to report context-specific data surrounding their medication intake behavior, self-reported pain, and vaso-occlusive crises, with the ultimate goal of providing a cost-effective approach for monitoring adherence and contextualizing self-reported pain.

## 2. Methods

### 2.1. App Development

The development of the designated OpPill Mobile Application was carried out to reflect the needs of the SCD community and allow for input related to recurring concerns gathered from surveys of focus groups of users. To be certain that the concerns of the patients, as well as clinicians, were addressed, comprehensive functional parameters were identified with the establishment of a focus group. This group engaged in dialogue through regular discussions to continue refining the functional requirements of the application prior to rating with the MARS tool. The overarching software requirements included (see Figure 1 for final app details):Easy-to-use and intuitive graphical interface at each layer for all users, such as SCD patients and clinical staffInclusive of functional parameters that allow effortless documentation of medication adherence, as well as symptomology, to collect data that are relevant to the clinical support team for SCDAll graphical interfaces present easy-to-identify options, such as for entry of medication adherence including drop down list options and medication imagesData transmission capabilities include meeting HIPAA requirements regarding privacy parametersAll data transmission and storage include secure practicesMulti-platform capability was included to ensure app usage across diverse mobile devices. Thus, the app was designed to be implemented for android and apple smartphone devices.

### 2.2. Data Collection

Data were collected from an outpatient ambulatory care clinic at the Virginia Commonwealth University (VCU) Medical Center, a large urban teaching hospital located in Richmond, Virginia. In this study, patients were introduced to the application during their clinic visit and were subsequently instructed to use the application until their next clinic visit where they would provide their rating of the application using the MARS tool. We collected participant demographics, including age, race, gender, education (highest completed), and self-reported income. Opioid adherence and disease characteristics were also collected in the form of phenotypic manifestation of pain (pain intensity, pain frequency, pain location, self-description of pain characteristics) and disease genotype. Acceptability, usability, and efficacy of the OpPill was tested using the validated Mobile Application Rating Scale (MARS) tool. The MARS tool was designed by a research team involved in the development and validation of eHealth and mHealth interventions, or ‘eTools’. The scale aimed to provide researchers, clinicians, and developers with a list of evaluation criteria and a gradient response scale for their objective evaluation. There are three main MARS factors: (1) the MARS mean: this is the mean of four objective subscales (Engagement, Functionality, Aesthetics, and Information); (2) Subjective Quality; and (3) Perceived Impact. Subjective Quality and Perceived Impact are based on the rater’s own impression of the eTool, including its usability and perceived effectiveness.

The subjective quality and application-specific scales were customized for SCD following the MARS guidelines and scored on a scale of 1–5, where 1 is strongly disagree and 5 is strongly agree. Additionally, alternative pain-coping practices, along with body temperature at the time of collection, were captured. All de-identified information was maintained in a HIPAA-compliant manner, and the study was IRB-approved.

### 2.3. Data Analysis

Patient demographics were reported using descriptive statistics. The application’s quality criteria clustered within the Engagement, Functionality, Aesthetics, and Information Quality were evaluated by assessing the mean and standard deviation for each category. The subjective qualities, customized to SCD, were evaluated using the Spearman correlation to assess the perceived impact of the application on the user’s knowledge, attitudes, intentions to change, as well as likelihood to actually change. A *p*-Value of less than 0.05 means that the correlation is statistically significant.

## 3. Results

### 3.1. Study Participants

Patients’ ages ranged from 18.77 to 58.83, with a mean of 36.56 years. Nearly half (46.4%, *n* = 13) of participants were male, and 53.6% (*n* = 15) were female. Data from two patients were withdrawn due to acute onset of vaso-occlusive crisis. In terms of participant education, 25% (*n* = 7) had completed a high school degree or a GED equivalent, 39.3% (*n* = 11) had completed some college, 14.3% (*n* = 4) had completed a degree equivalent to that of a 2-year college, 7.1% (*n* = 2) had completed a 4-year degree, and 14.3% (*n* = 4) had completed a master’s degree. None of the participants had completed doctoral or professional education. Income representation was spread from <$10,000 (25%), $10,000–$19,000 (29.2%), $20,000–$29,000 (12.5%), $40,000–$49,000 (12.5%), $50,000–$59,000 (4.2%), and ≥$60,000 (4.2%). Sickle cell patients’ self-reported genotypes were as follows: Hemoglobin SS 26.9% [7], Hemoglobin SC 42.3% [11], Hemoglobin S β0 Thalassemia 11.5% [3], Hemoglobin S β+ Thalassemia 3.8% [2], and 15.4% unsure.

### 3.2. MARS Tool Scores

The app quality criteria were divided as Aesthetics, Engagement, Functionality, Information Quality, subjective quality categories, and app/disease-specific scores. Each MARS item used a 5-point Likert-type scale (1—Inadequate, 2—Poor, 3—Acceptable, 4—Good, 5—Excellent). Results for the first four categories are summarized in Figure 2 and Figure 3 where means, medians, and outliers are reported.

(1)Engagement

Engagement was gauged by assessing the application’s ability to be fun, interesting, customizable, interactive (e.g., sends alerts, messages, reminders, feedback, enables sharing), and well-targeted to audience. Although not vigorously found to be entertaining, the overwhelming majority found the application to be well-targeted, interactive, and customizable. The average Engagement score was (M = 3.93, SD = 0.73).

(2)Functionality

The application’s Functionality was assessed by asking patients to report on the functioning of the app, ease to learn, navigation, flow logic, and gestural design. On this topic, results indicate that the majority of patients found the application to be easy-to-use and learn, to perform as intended, and easy-to-navigate. The average Engagement score was (M = 4.54, SD = 0.66).

(3)Aesthetics

Aesthetics of the application were assessed by asking patients to rate questions regarding the app’s graphic design, overall visual appeal, color scheme, and stylistic consistency. Although two people indicated that the application did not look good and one patient reported that the application had a bad design, the majority found the application’s layout to be satisfactory, clear, or professional. Most people found the application’s graphics to be of good quality and to have high visual appeal. The average aesthetic score was (M = 3.92, SD = 0.81).

(4)Information Quality

The application’s quality of information was rated by participants through questions asking them to rate the content for accuracy, quality, quantity, goals, and understanding. Mean scores indicate that the majority found the information to be of high quality. The average Information Quality score was (M = 3.91, SD = 0.87).

(5)Subjective Quality

The subjective quality scale asked the participant to rate whether they would recommend the application to other patients, how often they would use the application in a 12-month period if given the opportunity, whether they were willing to pay for the application, and what the overall rating of the application is. See Figure 4 for results.

(6)App-Specific—Sickle Cell Disease

The sixth category of the MARS Tool (App Specific) was modified to include questions related to sickle cell disease specifically. To gain an understanding of the factors impacting the ability of the application to improve the user’s awareness, knowledge, attitudes, intentions to change, likelihood to seek help, and ability to change behavior surrounding SCD, the Spearman rank correlation and ANOVA were used to evaluate the relationship between sickle cell-specific responses, subjective qualities, age, education level, genotype, income, and the MARS classifications (Engagement, functions, Aesthetics and Information Quality). These results are summarized in Table 1 and Figure 5.

The Spearman rank correlation calculations showed several positive relationships between objective scales (Engagement, Information Quality, Functionality, and Aesthetics) and sickle cell-specific questions. Most notable was the correlation between Engagement and the sickle cell-specific questions, as well as the correlation between the willingness to recommend the application and the MARS objective scales (R = 0.73, *p* = 0.0001). There was neither a statistical significance in the correlation of the application’s Functionality and most of the sickle cell-specific questions, nor a significant relationship between the demographics and any of the MARS categories.

Given the small sample size in our study, we dichotomized some of the demographic information: gender as male vs. female, education as high school and below vs. college and above, and income as ≤$25,000 vs. >$25,000 based on the federal poverty level. Genotype was not dichotomized because it is indicative of a specific diagnostic that cannot be grouped. Although income had an impact on how functionality was scored (mean difference = −0.5, with a higher mean for female, *p* = 0.05, *d* = 2.6). The Student’s t-test indicated that there was no statistical significance between the MARS sores and gender or education. This was further validated with small effect sizes between all the categories of the MARS scores and the demographics, except for income level, which influenced functionality scores (See Table 2).

## 4. Discussion

The goal of this study was to test the feasibility and quality of a mobile application (OpPill) for adherence to prescribed opioids by asking patients to rate the application using the validated MARS tool. We defined feasibility as being an average rating above 3 out of 5 for each of the subjective sickle cell-specific measures, and good quality as being an average above 3 out of 5 for each of the MARS tool categories (Engagement, Functionality, Aesthetics, and Information Quality), or overall application rating above 3 out of 5 from the subjective app quality scale. The MARS tool, a reliable, multidimensional measure for trialing, classifying, and rating the quality of mobile health apps was used to evaluate the quality of our application. The evaluation was divided into categories including: Engagement, Functionality, Aesthetics, and Information Quality. Each section was scored according to the MARS tool scoring guide by calculating the mean of each of the above-named scales and the mean app quality total score. The application’s Functionality rated highest (M = 4.54, SD = 0.66), followed by Engagement (M = 3.93, SD = 0.73), Aesthetics (M = 3.92, SD = 0.81), and Information Quality (M = 3.91, SD = 0.87). The mean app quality total score was M = 3.98, SD = 0.77. In the subjective section of the application, the average recommending score was 4.4, which implied that patients were very likely to recommend this app to others. When asked how many times they would use the application in a 12-month time frame, the average score was 3.8, indicating that patients will use it over 10 times in the next 12 months. Although the overall rating for this application was 3.5 out of 5, suggesting an application that is above-average, most of the patients indicated that they would not be willing to pay for such an application. This could be due to the socio-economical factors that plague sickle cell disease [27].

Although there are several studies evaluating the use of web-based applications in medicine, there is currently no published literature evaluating the acceptance and feasibility of web-based applications for adherence to prescribed opioids for adults with SCD. Therefore, building on the principle that mHealth can offer numerous accessible techniques to help patients take their medications, given their customizable content, affordability, and portability, we offer an insight and comparisons of our results with past research on non-cancer pain conditions.

Traditionally, SCD care has been centered around managing pain to acceptable levels. Some findings suggest that patients are more interested in an approach to pain management that would not only reduce the pain, but also allow them to easily integrate into society and the workplace with independence, despite the chronic condition. Many participants emphasized their need to perform (limited) activity while managing their disease. Care for SCD patients may best be optimized with better medication prescribing behavior, better healthcare delivery, and better overall support. This is not always the case due to the disparities affecting patients with SCD [28].

Our results were similar to those reported in a review and content analysis of Engagement, Functionality, Aesthetics, Information Quality, and change techniques in the most popular commercial apps for weight management using the MARS tool. Bardus et al. reported using the tool to independently assess 23 popular apps’ features, quality, and content. Their reported results are: Engagement (M = 3.0, SD = 0.9), Functionality (M = 3.8, SD = 0.9), Aesthetics (M = 3.4, SD = 1.2), and Information Quality (M = 2.2, SD = 0.7), with a total score (M = 3.1, SD = 0.8) [29]. An emerging trend with Functionality leading in rating scores is observed between our study and their review, although our overall performance per category is superior. These results reflect the quality of the application that was developed.

Correlation analyses assessed relations between the objective subscales (Engagement, Functionality, Aesthetics, and Information Quality) and sickle cell-specific questions. The app quality indicated by MARS scores was positively correlated with a number of sickle cell-specific topics. There were no significant relationships found between age, gender, and education and any of the MARS categories, suggesting no bias due to age or level of education in the application’s rating using the MARS tool. However, *t*-test and effect size indicated that income level played a role in how female patients rated the application’s functionality. Female patients with income below $25,000 rated the application’s functionality higher than their counterparts, perhaps indicating the need to focus future work on improving the application’s functionality. The only positive relationship associated with the willingness to pay for the application amongst the MARS categories was with Engagement (R = 053, *p* = 0.0023). This indicates that patients may be willing to pay for medical applications if they find the content to be engaging towards them or their medical condition. Another singular positive relationship was that of the frequency of usage (how likely patients were to use the application in a 12-month time frame if relevant to them) and the application’s Aesthetics (R = 057, *p* = 0.0040), perhaps indicating that a patient’s willingness to frequently use applications is tied to the application’s looks more so than its functionality, engagement, and information quality.

### Limitations

This study highlighted the feasibility of a mobile software application as a means of measuring adherence to therapy and providing context-specific information regarding SCD patients’ medication intake behavior and their self-reported pain. Limitations in this study can be attributed to the fact that this was a feasibility study with a small sample size. Findings could be biased according to the Hawthorne effect, which refers to the inclination of some people to work harder and perform better when they are being observed as part of an experiment. That is, in a clinical research environment, positive results could be due to the simple fact that participants are aware that they are being observed. Additionally, this initial study focused on evaluating the application’s feasibility and quality; future work should consider adding a verification mechanism to validate the self-reported entries. Potential verification mechanisms could be items such as the new FDA-approved digital pill, a medication embedded within an ingestible sensor that could provide objective verification of medication adherence.

## 5. Conclusions

The rapid pace of technological development provides great opportunity for more disease-oriented web-based applications to improve patient care and health outcomes. Although a large number of medication-related apps are available, the majority of them cover a broad spectrum of disease and lack specific focus for one disease process, potentially limiting disease-specific details and contextual information for SCD patients’ particular medication intake behavior. Initial testing of the OpPill application, as described herein, showed that the application ranked well when assessing for Engagement, Functionality, Aesthetics, and Information quality in the targeted population. Engagement scores were high, indicating that patients want to become more engaged in their own health care, and with patient-specific applications. Thus, we provide an opportunity to positively impact behaviors and improve adherence in adult patients with sickle cell disease. Future work could include a longitudinal randomized control trial in adults with sickle cell disease to assess whether patients using the application adhered to their medication better than the control group.

## Figures and Tables

**Figure 1 healthcare-10-01506-f001:**
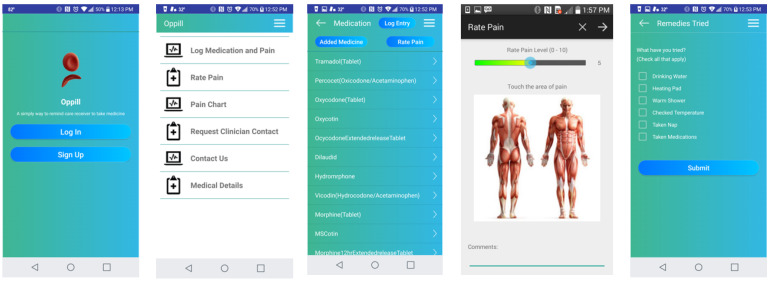
OpPill Application Screenshots.

**Figure 2 healthcare-10-01506-f002:**
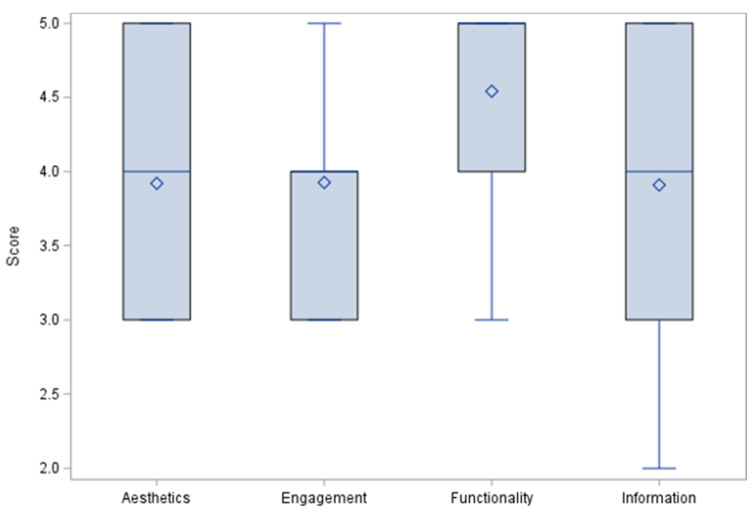
MARS tool category scores.

**Figure 3 healthcare-10-01506-f003:**
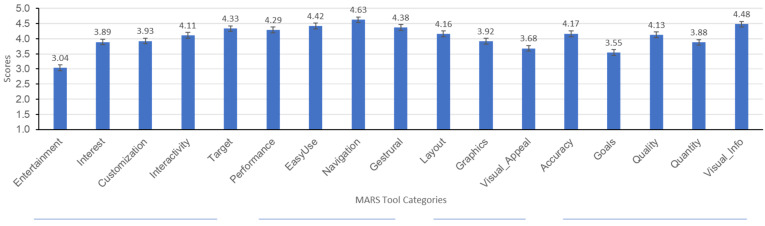
MARS Tool category distribution.

**Figure 4 healthcare-10-01506-f004:**
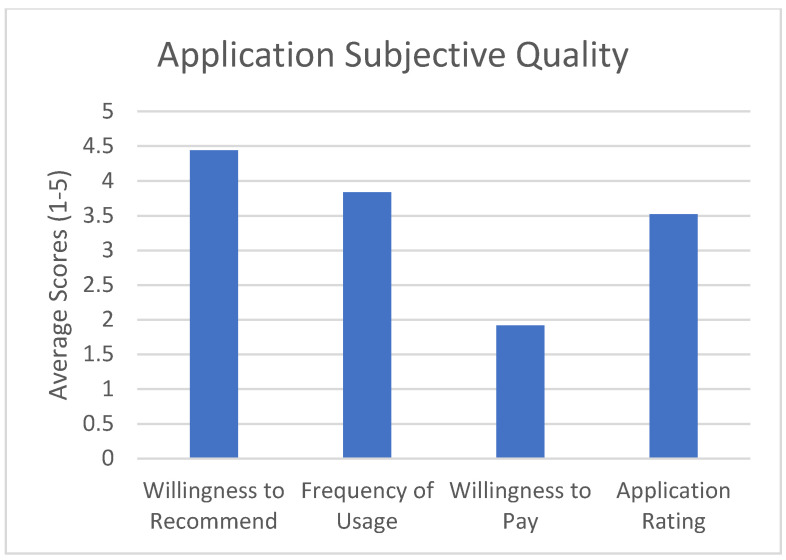
Application subjective quality ratings.

**Figure 5 healthcare-10-01506-f005:**
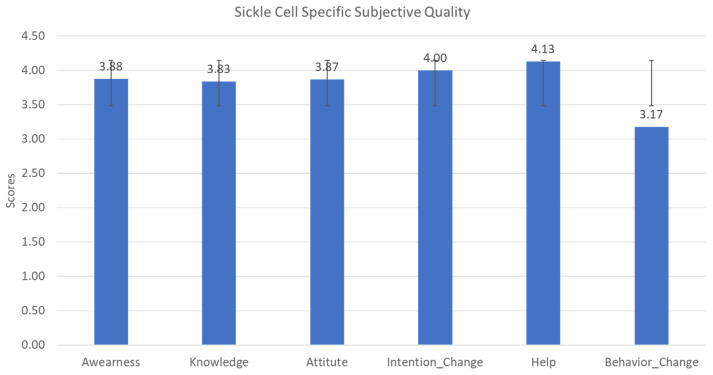
Sickle cell-specific application quality scale distribution.

**Table 1 healthcare-10-01506-t001:** Spearman Rank Correlation/ANOVA*—Application Impact.

Application Impact (Spearman Rank Correlation Coefficients/ANOVA* R-Square) (Prob > |r|)				
	Engagement	Functionality	Aesthetics	Information
Awareness	0.60 (0.0032) **	0.08 (0.7343)	0.45 (0.0281) **	0.54 (0.0110) **
Knowledge	0.56 (0.0057) **	0.0020 (0.9929)	0.34 (0.0989)	0.42 (0.0578)
Attitude	0.67 (0.0006) **	0.18 (0.4298)	0.46 (0.0288) **	0.47 (0.0385) **
Intention to Change	0.61 (0.0020) **	0.09 (0.6838)	0.46 (0.0237) **	0.57 (0.0075) **
Help Seeking	0.55 (0.0063) **	−0.04 (0.8458)	0.29 (0.1726)	0.29 (0.2011)
Behavior Change	0.54 (0.0101) **	0.21 (0.3380)	0.58 (0.0039) **	0.43 (0.0561)
Age	−0.15 (0.3310)	0.06 (0.9100)	−0.25 (0.0879)	−0.03 (0.8949)
Gender	0.008 (0.6454) *	0.000014 (0.9849) *	0.009 (0.6461) *	0.01 (0.6349) *
Education	0.24 (0.2566) *	0.07 (0.9083) *	0.20 (0.4575) *	0.19 (0.5726) *
Income	0.18 (0.5514) *	0.22 (0.4566) *	0.11 (0.8554) *	0.28 (0.4384) *
Genotype	0.40 (0.0588) *	0.28 (0.1343) *	0.39 (0.1075) *	0.27 (0.2868) *
Willingness to Recommend	0.41 (0.0425) **	0.73 (<0.0001) **	0.58 (0.0019) **	0.40 (0.0308) **
Frequency of Usage	0.08 (0.5634)	0.24 (0.1738)	0.57 (0.0040) **	0.41 (0.0711)
Willingness to Pay	0.53(0.0023) **	0.30 (0.1235)	0.23 (0.1781)	0.32 (0.1631)
Application Rating	0.73 (<0.0001) **	0.64 (0.0008) **	0.78 (<0.0001) **	0.73 (<0.0001) **

* ANOVA for Gender, Education, Income, and Genotype (Categorical variables). ** Indicates statistical significance.

**Table 2 healthcare-10-01506-t002:** Mean Difference (*p*-Value) and Effect Size of Each Application Category Between Gender, Education, and Income.

	Mean Diff. (|t|)	*p*-Value	Cohen’s d (Effect Size)
Gender (M–F)			
Engagement	−0.005	0.98	0.01
Functionality	−0.08	0.76	0.12
Aesthetics	0.15	0.65	0.18
Information	−0.18	0.64	0.20
Education (College and Above–High School and below)			
Engagement	−0.09	0.77	0.12
Functionality	0.07	0.81	0.11
Aesthetics	−0.15	0.73	0.18
Information	−0.14	0.75	0.16
Income (Above Poverty–Below Poverty)			
Engagement	−0.08	0.80	0.11
Functionality	−0.5	0.05 *	2.6
Aesthetics	0.09	0.80	0.12
Information	0.3	0.39	0.42

* Indicates Statistical significance.

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
