# Peer review of "Feasibility and Quality Validation of a Mobile Application for Enhancing Adherence to Opioids in Sickle Cell Disease"

_healthcare, 2022, doi:10.3390/healthcare10081506_

Round 1

Reviewer 1 Report

This is a nicely written manuscript that presents user feedback data on a new mHealth app for tracking pain and opioid use among adults with sickle cell disease (SCD). The presented app is not particularly novel as several groups have already published on pain and medication tracking apps in sickle cell (e.g. Palermo 2014 - Pain, Jacob 2013 - J Am Assoc Nurs Pract, Shah - Hemoglobin 2014, Crosby 2017 - Ped Blood Cancer), thus, it is not clear why the authors had to develop their own app rather than building upon prior technology and the literature from the past two decades. Even more surprising is that none of these prior studies of SCD self-management and monitoring apps were cited. 

The abstract and introduction begin by highlighting the importance of literacy in the U.S. and how low health literacy contributes to poor medication adherence. However, the study in no way addresses health literacy or literacy. Literacy (either general or health-focused) is not measured or discussed anywhere else in the manuscript. Please remove discussion of literacy from the manuscript or provide some data on literacy and how it may impact usability of the app. 

Minor point: 

ABSTRACT. Both the terms “health mobile apps” and “mobile health apps” are used. Keep the terminology consistent throughout. 

Author Response

Response to Reviewers

Reviewer 1: 

This is a nicely written manuscript that presents user feedback data on a new mHealth app for tracking pain and opioid use among adults with sickle cell disease (SCD). The presented app is not particularly novel as several groups have already published on pain and medication tracking apps in sickle cell (e.g. Palermo 2014 - Pain, Jacob 2013 - J Am Assoc Nurs Pract, Shah - Hemoglobin 2014, Crosby 2017 - Ped Blood Cancer), thus, it is not clear why the authors had to develop their own app rather than building upon prior technology and the literature from the past two decades. Even more surprising is that none of these prior studies of SCD self-management and monitoring apps were cited. 

We have included a section about the prior Applications that were developed although their work focused on children. This highlights the need for applications for adults with Sickle Cell disease as literature indicates that transition from pediatric to adult care in sickle cell disease typically fails and could lead to increase mortality due to disease severity, psychological and social factors.

The abstract and introduction begin by highlighting the importance of literacy in the U.S. and how low health literacy contributes to poor medication adherence. However, the study in no way addresses health literacy or literacy. Literacy (either general or health-focused) is not measured or discussed anywhere else in the manuscript. Please remove discussion of literacy from the manuscript or provide some data on literacy and how it may impact usability of the app. 

Although American health literacy is poor and typically translates into poor medication adherence, we removed this section from the paper given that we did not actually measure health literacy on this project.

Minor point: 

ABSTRACT. Both the terms “health mobile apps” and “mobile health apps” are used. Keep the terminology consistent throughout. 

This has been addressed. Thank you for the suggestion. 

Reviewer 2 Report

This manuscript described the development and assessment of the acceptability and usability of a mobile application for adult patients with Sickle Cell Disease designed to increase adherence to prescribed opioids. The manuscript was overall well-written and organized. It also addresses an important topic relevant to current prescription practices for patients whose pain is often not well managed in our health care system. Additional data in the Methods section and the Discussion section would increase understanding of how this application was tested and how it relates back to health care literacy and improved adherence.

Method

Page 4

Please provided additional information regarding participants’ use of the application.  Did they use the application in the Clinic or at home, and for how long did they try out the application?

How were the subjective qualities exactly measured in terms of user’s knowledge, attitudes, intention to change and likelihood to change? Was a specific measurement tool used or 4 specific questions, and how were the items or questions scored?

Results

Page 7   Reporting of correlations and significance levels would not be to be repeated in the text and in Table 1.

Discussion

Page 7   Health care literacy was introduced early in the Introduction and its relationship to adherence. It is not mentioned in the discussion so its relevance to this application may be lost.

Page 8   Perhaps another phrase then “quench the ache, could be used that validates the quality and quantity of pain experienced by patients with SCD.

Page 8/9               Suggestions for future research on how this application could be tested to verify improved adherence to prescribed opioids should be included in this section.

Page 9 Reference to bridging the gap of trust between patient and provider was made in the final sentence of the manuscript but this issue was not introduced or mentioned throughout the article?  Please provide additional references or transitions between all concepts introduced throughout the manuscript.

Author Response

Response to Reviewers

Reviewer 2: 

This manuscript described the development and assessment of the acceptability and usability of a mobile application for adult patients with Sickle Cell Disease designed to increase adherence to prescribed opioids. The manuscript was overall well-written and organized. It also addresses an important topic relevant to current prescription practices for patients whose pain is often not well managed in our health care system. Additional data in the Methods section and the Discussion section would increase understanding of how this application was tested and how it relates back to health care literacy and improved adherence.

Method

Page 4

Please provided additional information regarding participants’ use of the application.  Did they use the application in the Clinic or at home, and for how long did they try out the application?

How were the subjective qualities exactly measured in terms of user’s knowledge, attitudes, intention to change and likelihood to change? Was a specific measurement tool used or 4 specific questions, and how were the items or questions scored?

We have included more information regarding how the application was used prior to being ranked with the MARS tool. We added more results, and the tool was also added as an appendix to provide more information about the questions that were asked.

Results

Page 7   Reporting of correlations and significance levels would not be to be repeated in the text and in Table 1.

We have addressed this issue. Thank you for the recommendation.

Discussion

Page 7   Health care literacy was introduced early in the Introduction and its relationship to adherence. It is not mentioned in the discussion so its relevance to this application may be lost.

Although American health literacy is poor and typically translates into poor medication adherence, we removed this section from the paper given that we did not actually measure health literacy on this project.

Page 8   Perhaps another phrase then “quench the ache, could be used that validates the quality and quantity of pain experienced by patients with SCD.

This has been addressed, Thank you for the suggestion.

Page 8/9               Suggestions for future research on how this application could be tested to verify improved adherence to prescribed opioids should be included in this section.

We added a sentence addressing this issue. Thank you for the recommendation.

Page 9 Reference to bridging the gap of trust between patient and provider was made in the final sentence of the manuscript but this issue was not introduced or mentioned throughout the article?  Please provide additional references or transitions between all concepts introduced throughout the manuscript.

We realized that since this project was testing the quality and gathering feedback regarding the application, it was best to remove the topic regarding the gap of trust between patients and providers. However, this remains an issue in Sickle Cell Disease care that could be assessed in a randomized control trial. We do recognize that although Applications can objectify a patient’s behavior, patients could still falsely report through the application thus indicating the need for an objective validating tool not relying on patient entry, but rather on physiological parameters such as bio-ingestible pills.

Round 2

Reviewer 2 Report

This manuscript is a revised submission on the development and assessment of OpPill, a mobile application for adult patients with Sickle Cell Disease designed to increase adherence to prescribed opioids. The authors addressed all of this reviewer's previous questions and concerns. Below are 3 minor points that may be considered to ensure relevant findings are highlighted and statements are appropriately cited. 

Results

The description of Table 1 could highlight the most relevant findings: 1. significant correlations between Engagement and all SCD-specific questions, 2. no significant correlations with any demographic characteristics, and 3. significant correlations between Willingness to Recommend and all objective subscales.  These findings are then addressed in the Discussion section.

Discussion

Please provide citations for the last sentence in the first paragraph related to socio-economical factors that plague sickle cell disease and last sentence in the third paragraph related to disparities affecting patients with SCD.

Author Response

Reviewer 2:

This manuscript is a revised submission on the development and assessment of OpPill, a mobile application for adult patients with Sickle Cell Disease designed to increase adherence to prescribed opioids. The authors addressed all of this reviewer's previous questions and concerns. Below are 3 minor points that may be considered to ensure relevant findings are highlighted and statements are appropriately cited. 

Results

The description of Table 1 could highlight the most relevant findings: 1. significant correlations between Engagement and all SCD-specific questions, 2. no significant correlations with any demographic characteristics, and 3. significant correlations between Willingness to Recommend and all objective subscales.  These findings are then addressed in the Discussion section.

These modifications were made. Thank you for the recommendation.

Discussion

Please provide citations for the last sentence in the first paragraph related to socio-economical factors that plague sickle cell disease and last sentence in the third paragraph related to disparities affecting patients with SCD.

Thank you for these recommendations. The requested citations have been included.